# COVID-19 in Pregnancy in Scotland (COPS): protocol for an observational study using linked Scottish national data

Sarah Jane Stock [1,2] David McAllister [3,4] Eleftheria Vasileiou [2]
Colin R Simpson [5] Helen R Stagg [2] Utkarsh Agrawal [6]
Colin McCowan [6] Leanne Hopkins [7] Jack Donaghy [7] Lewis Ritchie [8]
Chris Robertson [9] Aziz Sheikh [2] Rachael Wood [2,7,10]

For numbered affiliations see end of article.

**Correspondence to**
Dr Sarah Jane Stock;
sarah.stock@ed.ac.uk

## ABSTRACT

**Introduction** The effects of SARS-CoV-2 in pregnancy are not fully delineated. We will describe the incidence of COVID-19 in pregnancy at population level in Scotland, in a prospective cohort study using linked data. We will determine associations between COVID-19 and adverse pregnancy, neonatal and maternal outcomes and the proportion of confirmed cases of SARS-CoV-2 infection in neonates associated with maternal COVID-19.

**Methods and analysis** Prospective cohort study using national linked data sets. We will include all women in Scotland, UK, who were pregnant on or became pregnant after, 1 March 2020 (the date of the first confirmed case of SARS-CoV-2 infection in Scotland) and all births in Scotland from 1 March 2020 onwards. Individual-level data will be extracted from data sets containing details of all livebirths, stillbirth, terminations of pregnancy and miscarriages and ectopic pregnancies treated in hospital or attending general practice. Records will be linked within the Early Pandemic Evaluation and Enhanced Surveillance of COVID-19 (EAVE II) platform, which includes primary care records, virology and serology results and details of COVID-19 Community Hubs and Assessment Centre contacts and deaths. We will perform analyses using definitions for confirmed, probable and possible COVID-19 and report serology results (where available). Outcomes will include congenital anomaly, miscarriage, stillbirth, termination of pregnancy, preterm birth, neonatal infection, severe maternal disease and maternal deaths. We will perform descriptive analyses and appropriate modelling, adjusting for demographic and pregnancy characteristics and the presence of comorbidities. The cohort will provide a platform for future studies of the effectiveness and safety of therapeutic interventions and immunisations for COVID-19 and their effects on childhood and developmental outcomes.

**Ethics and dissemination** COVID-19 in Pregnancy in Scotland is a substudy of EAVE II(, which has approval from the National Research Ethics Service Committee. Findings will be reported to Scottish Government, Public Health Scotland and published in peer-reviewed journals.

### Strengths and limitations of this study

► We will interrogate Scottish national data at the population level to provide information on the incidence of, and outcomes following, COVID-19 outcomes in pregnant women.
► We are expanding an existing national pandemic reporting platform (Early Pandemic Evaluation and Enhanced Surveillance of COVID-19 EAVE II) to include assessment of all pregnancy outcomes. EAVE II uses deidentified individual patient-level data for almost the entire population of Scotland from general practices, hospitals, death registry, virology (reverse transcriptase PCR) and serology tests to investigate the epidemiology of COVID-19.
► This is an observational study and residual confounding is a potential concern.

## INTRODUCTION

The effects of novel SARS-CoV-2 in pregnancy are yet to be fully delineated.[1] Pregnant women are at greater risk of complications and severe disease from infection with other coronaviruses including SARS and middle eastern respiratory syndrome.[2][3] Pregnant women were thus identified as a potential vulnerable group in some countries and advised to take additional precautions as the COVID-19 pandemic unfolded.[2–4]

To inform public health policy, it is crucial to determine the effects of SARS-CoV-2 infection on maternal, pregnancy and neonatal health. SARS-CoV-2 transmission from mother to baby (antenatally or intrapartum) appears to be possible,[5] but the proportion of pregnancies affected and the clinical significance are uncertain. Potential effects of the virus on miscarriage, congenital anomalies, fetal growth, timing of delivery and stillbirth are unknown. We know from other viral

infections in pregnancy that infections with mild maternal symptomatology can have substantial impacts on the developing fetus (eg, cytomegalovirus, parvovirus, zika virus),[6] although mechanisms of placental transmission of virus vary.[7] The canonical receptors for SARS-CoV-2 (the angiotensin-converting enzyme 2 receptor and the serine protease TMPRSS2) are not coexpressed in the placenta, making placental infection unlikely.[8 9] Nevertheless, case reports have shown evidence suggestive of viral infection of the placenta, in association with pregnancy complications such as pre-eclampsia and abruption[10] and second trimester miscarriage.[11] Reports that include neonatal test results for SARS-CoV-2 show positive cases only in a minority of babies, with significant respiratory disease being rare in neonates.[12] However, some babies born to mothers who had COVID-19 have increased concentrations of both immunoglobulin IgM and IgG for SARS-CoV-2.[13 14] As IgM cannot cross the placenta, neonatal circulating SARS-CoV-2 IgM indicates vertical transmission of virus, although all the infants in reports so far have been asymptomatic and tested negative for SARS-CoV-2 viral RNA at birth.[13 14] There are also plausible links between SARS-CoV-2 and pregnancy complications such as preterm birth, which may be mediated either as a manifestation of COVID-19 disease itself or indirectly through increased stress due to the pandemic and containment measures or through altered physician threshold for iatrogenic preterm delivery in women with infection.[1]

Understanding the effects of COVD-19 at different stages in pregnancy and perinatally will help inform policy on shielding strategies and advice to pregnant women and those considering pregnancy. It is also essential to inform immunisation strategies when vaccines are available. For example, immunisation in early pregnancy may help protect against maternal infection during pregnancy and reduce complications; but immunisation in later pregnancy may be preferential to provide passive immunisation to babies if neonatal infection is the predominant concern.

There are a number of surveillance studies gathering data on pregnant women with COVID-19 currently underway in the UK, summarised in table 1.

Collectively, these surveillance studies can provide detailed characterisation of selected groups of pregnant women (and neonates) affected by COVID-19. The study outlined in this protocol will complement these existing studies by providing population-based information (for the whole of Scotland) on the risks of, and outcomes following, COVID-19 at any stage of pregnancy for women in the community and/or admitted to hospital.

The primary objectives of the COVID-19 in Pregnancy in Scotland study are to:

a. Describe the incidence of SARS-CoV-2 infection and COVID-19, in the pregnant population.
b. Determine associations between COVID-19 and adverse maternal, pregnancy and neonatal outcomes.

c. Determine the proportion of neonates with confirmed SARS-CoV-2 infections that are associated with COVID-19 in the baby's mother

Secondary objectives are to:

a. Assess the proportion of COVID-19 cases in pregnant women and neonates who are included in relevant other enhanced surveillance studies (eg, British Paediatric Surveillance Unit (BPSU), Clinical Characterisation Protocol Tier 0 study (CO-CIN))[15–17].
b. Provide a platform to assess the safety and effectiveness of any new or existing prophylactic or therapeutic interventions (eg, new or repurposed therapies, vaccines, antimicrobials) in pregnant women and their babies.
c. Enable evaluation of the longer term sequelae of maternal SARS-CoV-2 and therapeutic interventions to mitigate SARS-CoV-2 in pregnancy and in children.

This COVID-19 in Pregnancy in Scotland (COPS) study is a substudy of the Early Pandemic Evaluation and Enhanced Surveillance of COVID-19 (EAVE II).[18–20] EAVE II is a national, real-time, data platform to identify the population groups most at risk from SARS-CoV-2 infection and COVID-19 disease and mortality, linking Scottish general practice (GP) records (5.4 million registered patients) with secondary care and laboratory datasets.[18] Pregnancy will be assessed as one of these at-risk groups. Within this COPS study protocol, we specify in detail the national data sets that will be incorporated within the EAVE II platform to enable pregnant women (and associated pregnancy start and end dates) to be reliably identified. We also specify in detail the maternal, pregnancy and neonatal outcomes following maternal COVID-19 that will be examined.

## METHODS

### Patient and public involvement

Parents and pregnant women have not been involved in design of this protocol. However, we will work in partnership with a patient and public involvement group set up for the EAVE II study regarding interpretation of results, presentation and dissemination of findings. We also have close links with Tommy's charity who will codevelop dissemination plans and help ensure that findings reach relevant stakeholders.

### Study design and population

This is a prospective cohort study using national maternity, community, hospital and laboratory-linked data sets, in Scotland, UK. We will include all women in Scotland who were pregnant on or became pregnant after 1 March 2020 (the date of the first confirmed case of SARS-CoV-2 infection in Scotland)[21] and all live-born babies born in Scotland from 1 March 2020 onwards. The end date for the study will be determined by the future development, and, in particular, suppression of the pandemic in Scotland.

Women in the cohort with the earliest dates of conception will only have been at risk of COVID-19 at the very

**Table 1** UK surveillance studies on COVID-19 in pregnant women and their babies

| Name of study | Institution | Inclusion | Reporting by | Consent required | Likely coverage in Scotland |
|---|---|---|---|---|---|
| COVID-19 in Pregnancy[47] | UK Obstetric Surveillance System study | Any women admitted to hospital in the UK with confirmed COVID-19 at any stage of pregnancy | Front-line clinicians | No | High |
| Pregnancy And Neonatal outcomes for women with COVID-19[48] | National Institute of Healthcare Research Imperial Biomedical Research Centre | Women who have suspected or confirmed COVID-19 at any stage during pregnancy and their babies | Front-line clinicians | Yes | Unknown as yet |
| Clinical Characterisation Protocol Tier 0 study[17] | The International Severe Acute Respiratory and emerging Infection Consortium | Any patient admitted participating hospitals in the UK with confirmed COVID-19 | Reporting is by research nurses | No | Low but may increase |
| Neonatal complications of COVID-19[17] | British Paediatric Surveillance Unit | All babies born to mothers with COVID-19 who are admitted to neonatal care (whether the baby has COVID-19 or not) and all babies with confirmed COVID-19 in the neonatal period. | Front-line clinicians. | No | High |
| Multisystem inflammatory syndrome, Kawasaki disease and toxic shock syndrome[15] | British Paediatric Surveillance Unit | All children less than 16 years old (including neonates) with multisystem inflammatory syndrome due to SARS-CoV-2 infection or otherwise unexplained. | Front-line clinicians | No | High |
| Understanding COVID-19 infection in women and their babies (periCOVID)[49] | Public Health England and St George's University London | Any pregnant woman with confirmed COVID-19 infection from 24 weeks gestation in England | Clinicians/research midwives and nurses | Yes | None |

end of their pregnancy. Women with more recent dates of conception will have been at risk for longer, up to women with date of conception from 1 March 2020 onwards, who will be at risk from conception onwards (until viral transmission is completely suppressed).

We aim to use a dynamic cohort of the entire pregnant population; therefore, selection bias is not anticipated and the data set will be fully generalisable to Scotland (with extensive generalisability to other high-income nations).

### Databases

Individual-level data will be extracted from the datasets listed below. Records relating to the same individual will be linked deterministically using the Community Health Index (CHI) number.[22] The CHI number is a unique identifier provided by National Health Service (NHS) Scotland for each resident registered with a GP. Public

Health Scotland (PHS) routinely adds both maternal and baby CHI numbers to live birth registration records: these 'spine' records will be used to facilitate intergenerational linkage of records relating to mothers and their babies.

An overall scheme for the planned linkage is provided in figure 1. Unless otherwise specified, included data sets are held by PHS and PHS is the data controller.

### Data sets to identify pregnant women in the general population and associated pregnancy start and end dates and pregnancy and neonatal outcomes

1. Antenatal booking records: this is a new national data return developed as part of the response to the COVID-19 pandemic providing information on all women booking for antenatal care with NHS maternity services throughout Scotland. It will be used for identification of women with ongoing pregnancies in near real time. All other records identify end of pregnancy

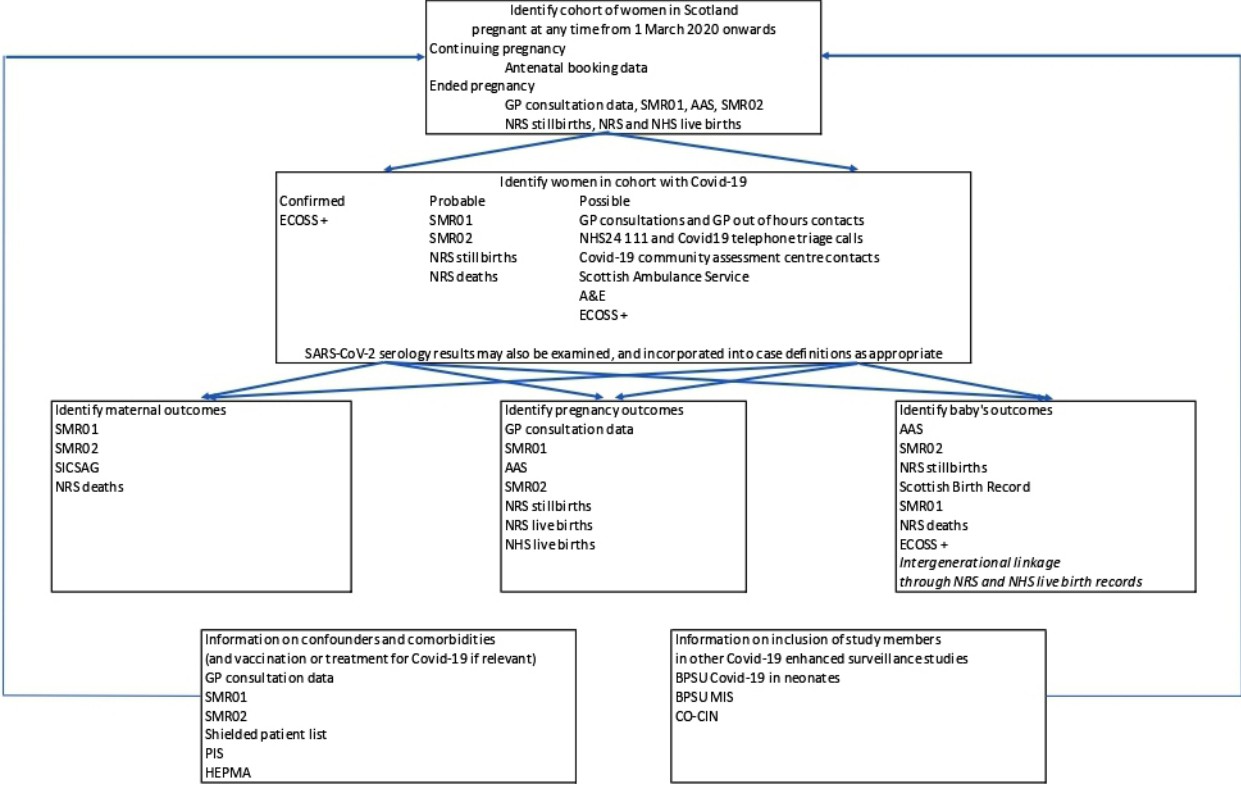

**Figure 1** Overview of data linkage for the COPS study. AAS,Abortion Act Scotland; BPSU, British Paediatric Surveillance Unit; CO-CIN,Clinical Characterisation Protocol Tier 0 study; COPS, COVID-19 in Pregnancy in Scotland; ECOSS,Electronic Communication of Surveillance in Scotland; GP, generalpractice;HEPMA, Hospital Electronic Prescribing and Medicines Administration; NHS, NationalHealth Service; NRS, National Records of Scotland; PIS, prescribing informationsystem;SMR, Scottish Morbidity Record.

events, thus, there is a time-lag before records are generated. More than 99% of births in Scotland book for antenatal care with NHS maternity services.[23]

2. Abortion Act Scotland (AAS) records: these are statutory notifications of termination of pregnancy and will be used to identify all terminations of pregnancy, including terminations of pregnancy indicated by congenital anomaly.[24]

3. National Records of Scotland (NRS) statutory stillbirth registrations: Scottish legislation requires all stillbirths at 24 weeks gestation or more to be registered with NRS.[25]

4. NRS statutory live birth registrations: Scottish legislation requires all live births at any gestation to be registered with NRS.[26]

5. NHS Scottish health board live births: A new national data return developed as part of the response to the COVID-19 pandemic (specifically to mitigate unavailability of NRS statutory live birth registration records when registration processes were suspended) providing information on all live births notified by maternity services to NHS Board child health administrative departments.

6. GP data: data from all patients registered in GPs are included in the EAVE II platform.[18] In COPS, these records will be used for identification of women with early miscarriage or ectopic pregnancy not

managed in hospitals and potential confounding comorbidities.

7. Scottish Morbidity Record (SMR) 01: the SMR01 database includes all general day case and in-patient admissions in Scotland.[27] Admissions to neonatal, maternity and mental healthcare are excluded from SMR01 as they are covered by other specialist data sets. SMR01 records are included in the EAVE II platform, and COPS will be used for identification of women with early miscarriage or ectopic pregnancy managed in hospitals and potential confounding comorbidities.

8. SMR 02: the SMR02 database includes all day case and in-patient admissions to maternity specialities in Scotland.[27] It will be used for identification of later miscarriage, stillbirth and live births managed in maternity units (≥98% of births in Scotland) and some home births (≤2% of births in Scotland).[23]

9. Scottish Birth Record (SBR): the SBR records basic demographic data on all births in Scotland and additional clinical information and diagnostic and operational procedure codes on babies admitted to neonatal care.[28] It will be used to identify neonates admitted to neonatal care.

10. Scottish Intensive Care Society Audit Group records: this is a national database of patients admitted to adult general critical care units in Scotland detailing

information on the management of critically ill or injured patients. All general intensive care units and combined ICU/high dependency units (HDUs) collect data and more than 90% of general HDUs and a number of specialist ICU and HDUs also provide records.[29] In COPS, these will be used to identify women admitted to critical care with COVID-19.

### Data sets to identify women with confirmed SARS-CoV-2 infection or COVID-19

1. Electronic Communication of Surveillance in Scotland (ECOSS)[30] and other viral reverse transcriptase PCR (RT-PCR) and serology results held separately by PHS are included on the EAVE II platform.[18] In COPS, these will be used for identification of pregnant women and neonates with positive viral RT-PCR and serology and women with negative viral RT-PCR.

   ECOSS is a database that holds surveillance data on various microorganisms (eg, influenza virus, coronavirus) and infections reported from NHS diagnostic and reference laboratories.[30] Data on laboratory results for all SARS-CoV-2 RT-PCR tests carried out in Scotland are being collated by ECOSS and can be linked to other data sources.[30]

   In substudies, residual sera from routine antenatal booking blood tests and 28-week gestation blood group and red cell antibody screen samples will be tested for SARS-CoV-2 antibodies. Residual sera from other blood tests conducted as part of routine (not COVID-19 related) primary and secondary care, and blood donation, are also being tested for SARS-CoV-2 antibodies as part of the surveillance of the pandemic in Scotland, and any results relating to pregnant women will also be incorporated.[31] Results from these substudies will be linked to pregnancy records and used to determine exposure to SARS-CoV-2 by the presence of antibodies.[18]

2. GP consultations, GP out of hours attendances, NHS24 calls, COVID-19 phone assessment hub calls and COVID-19 clinical assessment centre attendances are linked on the EAVE II platform.[18] We will extract data on pregnant women with possible COVID-19 from GP records and a network of COVID-19 Community Hubs and Assessment Centres established by NHS Health Boards across Scotland. These provide a direct and rapid route for people with COVID-19 symptoms that have worsened or not improved after a week to seek advice and primary care. The pathway for management of patients in the community with symptoms suggestive of COVID-19 has evolved as the pandemic has progressed. In early March 2020, patients with symptoms were advised to phone their GP in hours or NHS 24 out of hours for advice. Patients requiring face to face assessment were then seen in GP surgeries or GP out of hour centres. On 17 March 2020, the Scottish Government published details of a new national patient pathway, whereby all patients with symptoms (in or out of hours) were encouraged call NHS 24 as

the initial point of contact.[32] Patients thought likely to have COVID-19 were then passed to a new, dedicated NHS24 COVID-19 phone assessment hub. Those requiring face-to-face assessment in general were then seen in (or visited at home by staff from) an NHS24 COVID-19 clinical assessment centre, although pregnant women could alternatively be directed to their maternity service triage base.[32]

3. Unscheduled Care Datamart: This links data from NHS 24, Scottish Ambulance Service, Out of Hours Primary Care, Emergency Department, Acute and Mental Health admissions and Deaths to show a Continuous Unscheduled Care Pathway. Data are included in the EAVE II platform.[18] In COPS, we will use Scottish Ambulance Service incident and Accident and Emergency (A&E) attendance records for identification of women with possible COVID-19.[33]

4. NRS statutory death registrations will be used for identification of any women with COVID-19 recorded as cause of death.[34]

5. SMR01, SMR02 and NRS stillbirths will be used to identify women with COVID-19 recorded as cause of admission/stillbirth.[25 27]

### Data sets recording treatments, vaccination, shielding status and inclusion in other studies

1. Prescribing information system includes information on all prescribed medications that are dispensed in the community in Scotland.[35] Lookback records are included in the EAVE II platform[18] and used to provide information on the presence of comorbidities. COPS records will also be used to provide information on COVID-19 treatments given.

2. Scottish Hospital Electronic Prescribing and Medicines Administration systems are currently available within four Scottish hospitals, and data from these are linked in the EAVE II platform[18] to provide data on medications for COVID-19 administered in hospitals.[36]

3. GP consultation records included on the EAVE II platform[18] will be used to extract information on vaccinations for SARS-CoV-2 and other viruses.

4. We will identify pregnant women and neonates with COVID-19 in existing enhanced surveillance studies using minimal 'flag' variables from the BPSU and CO-CIN studies[15–17] as well as other trials who can provide this data.

### Exposure and outcome definitions
#### Pregnancy, start and end dates and pregnancy outcomes

A pregnancy will be defined by presence of a record pertaining to a pregnancy. As well as identifying completed pregnancies, through a record of any pregnancy outcome, we will be able to identify ongoing pregnancies at population level, through inclusion of records from women booked for antenatal care.

The following definitions will be used for pregnancy outcomes. The codes and data sources used to identify

relevant records are described in detail in online supplemental material.

1. Ectopic pregnancy: this will include any early pregnancy loss where the pregnancy is implanted outwith the uterus
2. Spontaneous pregnancy losses: these will be defined as miscarriage at less than 20 weeks gestation, late fetal losses at 20–23 weeks if there are no signs of life; and stillbirth if birth occurs at 24 weeks' gestation or more and there are no signs of life. If numbers allow, the miscarriages will be further split into the more clinically meaningful outcome categories of miscarriage at less than 14 weeks gestation and 14 weeks gestation and beyond.
3. Termination of pregnancy: these will be subclassified by the grounds for termination of pregnancy of the Abortion Act 1967.[37]
4. Live birth: the birth of a baby at any gestation with signs of life. No lower gestational limit will be used although in practice around 22 weeks gestation would be considered the lower limit at which live-born babies may survive

Pregnancy start date will be taken as the date of conception. In pregnancies that have ended date of conception will be imputed from

'date of conception=pregnancy end date — (the number of weeks of gestation at pregnancy end+2 weeks)'.

In ongoing pregnancies that have booked with maternity services and have a documented estimated date of delivery (EDD), date of conception will be imputed from

'date of conception=date of antenatal booking—(gestation at booking+2 week)'.

In ongoing pregnancies without a documented EDD, date of conception will be imputed from

'date of conception=date of last menstrual period+2 weeks'.

It is standard care for women who book with maternity services in Scotland have a first trimester ultrasound scan (usually 11–13+6 weeks gestation) to determine EDD, from which gestation is calculated. Booking takes place around 10 weeks gestation.

### SARS-CoV-2 infection and COVID-19

We will use the following definitions for COVID-19.

► Confirmed COVID-19 in pregnancy will be defined as positive viral PCR for SARS-CoV-2 on a test.
► Probable COVID-19 will be defined as COVID-19 recorded on a hospital admission, stillbirth or a maternal death record (using International Classification of Diseases (ICD) 10 codes U07.1, U07.2, B34.2, B97.2).
► Possible COVID-19 will be defined as meeting one or more of the following criteria:
  – GP consultation or GP out of hours attendance coded as possible COVID-19.
  – NHS24 call coded as possible COVID-19.
  – Patient triaged to NHS24 COVID-19 phone assessment hub or clinical assessment centre.

  – Scottish Ambulance Service call for possible COVID-19.
  – A&E attendance coded as possible COVID-19.
  – Negative SARS-CoV-2 viral PCR test when the test was taken for clinical indications (ie, excluding tests taken for routine testing of asymptomatic individuals).

The above definitions are hierarchical, for example, a positive SARS-CoV-2 nucleic acid test assigns a woman to the confirmed COVID-19 group, regardless of the presence of other records.

We will use the indication for SARS-CoV-2 nucleic acid test, which is recorded in ECOSS records, to distinguish between tests taken for clinical indications and routine testing of asymptomatic individuals. If we can not distinguish between these groups, we will exclude women testing negative from both the 'case' and 'control' groups—and base our definition of possible case on presentation to various healthcare settings with relevant symptoms as described above.

SARS-CoV-2 infections during pregnancy will be identified if the event of interest (eg, SARS-CoV-2 nucleic acid test taken, admission, unscheduled care attendance) occurs between 14 days prior to the estimated date of conception (to include women who could be viraemic periconceptually) and the end of pregnancy.

We will seek to access serological data as these become available. We will report the proportion of women with circulating IgG and/or IgM for SARS-CoV-2 and may incorporate serology results in case definitions and/or use in additional analyses as data mature. The timing of exposure to/infection with SARS-CoV-2 is more difficult to ascertain from serology results than from the other indicators of (possible) infection listed above. Dates of serological testing; start and end of pregnancy and plausible start (and, if applicable, end) of transmission of SARS-CoV-2 in Scotland will be taken into account when identifying women with exposure before, during and after pregnancy. Seroconversion windows will also be considered for women with sequential serology results.

We have included hierarchical definitions of COVID-19 to mitigate against potential biases that may result from (1) limitations of diagnostic strategy performance and (2) variation in availability of diagnostic tests. The definition of confirmed COVID-19 (women who test positive on PCR) may potentially bias results away from the null hypothesis (due to false-negative PCR results). The definition of possible COVID-19 may potentially bias results towards the null hypothesis (due to false-positive 'diagnoses'). We will test our hypotheses in this observational cohort across the range of assumptions allowed by hierarchical definitions. Other diagnostic and exposure categories may be added as the pandemic develops and diagnostic criteria change.

### Fetal and neonatal outcomes

The following fetal and neonatal outcomes will be included.

1. Congenital anomaly (major structural anomaly as defined by European network of population-based registries for the epidemiological surveillance of congenital anomalies (EUROCAT)[38] diagnosed in any pregnancy terminated at any gestation due to anomaly; miscarriage or stillbirth at ≥20 weeks or live-born baby diagnosed at <28 days of age).
2. Preterm birth (<37 weeks) categorised as spontaneous or medically indicated (ie, following induction of labour or elective caesarean section undertaken to mitigate clinical risk).
3. Small for gestational age (birthweight <10th centile by WHO-UK90 growth reference.[39]
4. Admission to neonatal care.
5. Neonatal SARS-CoV-2 infection (currently defined as positive viral RT-PCR test on sample taken from baby aged 0–27 days, definition may be expanded to include results of serology tests as evidence and testing options accumulate).
6. Neonatal mortality (death of a live born baby at <28 days of age).
7. Extended perinatal mortality (stillbirth or neonatal mortality).

### Maternal outcomes

We will collect data on the following maternal outcomes:

► COVID-19 disease requiring any hospital admission (defined as a patient admitted within 14 days of confirmed or probable COVID-19 or with confirmed or probable COVID-19 during admission).
► Severe COVID-19 disease requiring critical care admission or resulting in death (defined as patient admitted to critical care or dying within 28 days of confirmed or probable COVID-19 or with confirmed or probable COVID-19 during hospital admission, regardless of recorded cause of death).
► Any maternal death (defined as the death of a woman while pregnant or within 42 days of the termination of pregnancy, irrespective of the duration and site of the pregnancy, from any cause related to or aggravated by the pregnancy or its management but not from accidental or incidental causes).

### Population characteristics and confounding factors

A number of maternal and pregnancy characteristics will be collected that could be potential confounders or effect modifiers.

1. Demographics including age band and socioeconomic status determined by the Scottish Index of Multiple Deprivation (SIMD) classification of material deprivation.[40] SIMD quintiles 1–5 refer to the small geographical areas (data zones) each containing 20% of the Scottish population, with quintile 1 indicating the most deprived areas. The SIMD is a combination of 38 indicators of the following seven domains: income, employment, health, education, housing, geographical access to services and crime. We will also include urban/rural status of maternal residence based on the urban/rural eightfold classification,[41] where 1 is assigned to large urban areas and 8 is assigned to remote rural areas. We recognise ethnicity to be a complex indicator variable related to sociodemographic factors, health systems use, pregnancy and health outcomes and genetics. We will explore the possibility of including self-reported maternal ethnicity although missing data may preclude this.

2. Clinical at-risk groups of individuals with certain underlying medical conditions thought to increase risk of COVID-related complications. The following clinical at-risk conditions are identified in the EAVE II platform[18]: (a) chronic respiratory disease, (b) chronic heart disease, (c) chronic liver disease, (d) chronic kidney disease, (e) chronic liver disease, (f) chronic neurological disease, (g) diabetes, (h) conditions or medications causing impaired immune function, (i) asplenia or dysfunction of spleen and (j) body mass index (BMI). In addition, we will link pregnancy records with the shielded patient list included in the EAVE II platform (those with extremely high risk of severe manifestation of SARS-CoV-2 infection, and hence advised by Scottish Government to 'shield' during the pandemic). We will recategorise clinical risks for the pregnant population as (1) diabetes (type I; type II; other prepregnancy; gestational diabetes), (2) clinically vulnerable risk group (for whom seasonal influenza vaccination is recommended outwith pregnancy), (3) clinically extremely vulnerable risk group (those advised to 'shield' during the pandemic)[42] and (4) no clinical at-risk condition. BMI, which can be associated with adverse pregnancy outcomes as well as COVID disease will be included separately, with prepregnancy BMI or BMI at antenatal booking categorised as underweight, normal, overweight, obese and severely obese according to WHO definitions.[43] Other categorisation will be considered depending on number of pregnant women with these conditions and emergence of patterns of risk for COVID-19 disease.

3. Smoking status in pregnancy will be presented into the following four categories: Current smoker, non-smoker, ex-smoker and not recorded. Smoking status will be taken from booking record and/or GP records.

4. Obstetric characteristics (SMR02) will include previous pregnancies, plurality (number of babies), drug and alcohol use, antenatal steroid administration, mode of birth or management of pregnancy loss.

5. Clustering of outcomes by the 14 different Maternal NHS Board areas of residence.

### Statistical analysis
#### General approach

Baseline characteristics of all study participants will be described in relation to presence and absence of confirmed, probable or possible COVID-19 and outcomes of interest. Mean, median and proportions, together with a measure of dispersion, will be provided where appropriate to describe differences between the various groups

of interest based on the nature of each variable. Missing data will be provided for each variable. Two-tailed hypothesis tests will be used for all study's outcomes, with 95% CIs presented to show precision of estimates, and p values will be reported. All analyses will be carried out using the R statistical programming language. We do not propose to make any formal statistical adjustment for the multiple comparisons as the principal aim of the study is to estimate the effect of COVID-19 infection on pregnancy outcomes. The estimated effects and 95% CIs will be reported for the range of outcomes. However, a caveat will be clearly expressed regarding the dangers of overinterpreting these data, given the multiple outcomes used, particularly if it transpires that conflicting results are obtained from the differing outcome measures. The approach to imputing estimated date of conception when gestation is missing on records indicating pregnancy status is detailed in online supplemental material 1. Missing data are otherwise not anticipated to be a substantial problem (and hence imputation techniques are not anticipated) but this will be confirmed once initial data extracts are available, and our approach to handling missing data will be confirmed prior to analysis.

Analyses will be updated monthly, providing results for sequential months, and also information on the cumulative risk of COVID-19 as women progress through their pregnancies. Simple smoothing techniques such as rolling averages will be used to facilitate presentation and interpretation of findings. We will also present our results as proportions of COVID-19 infection, together with CIs based on the Wilson method. We will describe the temporal changes in the proportion using cumulative risk models. Covariates such as the trimester of pregnancy, age of the mother and deprivation will also be included with a view to estimating the potential effects of these variables on the risk of COVID-19 infection

If/when numbers of cases allow, we will examine incidence of COVID-19, and report outcomes, in subgroups including by maternal age band, SIMD deprivation quintile, maternal NHS Board area of residence and maternal comorbidity status. We may assess whether findings are robust to more stringent or emerging definitions of confirmed and suspected infection, in sensitivity analyses. Other sensitivity and subgroup analyses may be indicated by initial findings. We will clearly state which analyses were prespecified and which were post hoc.

### Incidence of SARS-CoV-2 and COVID-19 in the pregnant population

We will perform descriptive analysis of the number of cases over the total number of pregnancies, that is, how many pregnant women have had confirmed, probable or possible COVID-19/total number of pregnant women. Where timing of infection is known, we will describe incidence of SARS-CoV-2 infection by trimester of exposure— first trimester (0–13 weeks gestation), second trimester (14–27 weeks gestation), third trimester (≥28 weeks gestation)[44]; with denominators consisting of ongoing pregnancies in each trimester.

### Associations between COVID-19 and adverse pregnancy, neonatal and maternal outcomes

Initially we will perform a descriptive analysis comparing pregnancy outcomes in women with and without (i) confirmed, (2) probable and (3) possible COVID-19. In order to create appropriate comparison groups, for each woman with COVID-19, we will identify 10 women without COVID-19, with an ongoing pregnancy matched on gestation of diagnosis, and additionally matched on maternal age and maternal deprivation level. We will explore the need to match additional parameters such as NHS Board.

Occurrence of the outcomes of interest will be compared in women with and without COVID-19 using simple descriptive statistics (eg, 95% CI for the difference in proportions, generated using methods that accommodate proportions close to zero) and visualised appropriately.

If/when sufficient cases of COVID-19 among pregnant women accrue, and the univariable comparisons described above suggest that outcomes differ between women with and without SARS-2-CoV or COVID-19, formal modelling will be undertaken to quantify the impact of infection on outcomes, adjusting appropriately for confounding. We will use direct acyclical graphs to identify which factors to adjust for to mitigate for confounding. Appropriate methods that accommodate the competing risk and time to event nature of pregnancy outcomes (example event history analysis and/or multistate modelling) will be used.

### Association of SARS-CoV-2 in neonates with maternal COVID-19

We will use summary statistics to describe neonatal SARS-CoV-2 (currently defined as positive viral PCR for SARS-CoV-2 on sample taken from a baby aged 0–27 days old) by presentation of COVID-19 in the mother in different time periods (apparent onset of maternal illness >14 days prior to birth; 14 days prior to birth—date of birth; day 1–13 following birth; day 14–27 following birth).

### Proportion of pregnant women and neonates with COVID-19 or SARS-CoV-2 that are included in relevant other enhanced surveillance studies (BPSU, CO-CIN)

We will use summary statistics to describe the number and proportion of cases included in the external surveillance studies, and any factors associated with inclusion, for example, hospital admission status and NHS Board area of residence.

### Creation of a platform to assess the safety and effectiveness of any new or existing prophylactic or therapeutic interventions and assessment of childhood outcomes after pregnancy exposure to COVID-19

We will use summary statistics to describe the treatments and prophylactic interventions used in pregnancy. We plan future linkage of data within COPS with child health and education data to allow assessment of long-term outcomes.

**Table 2** Estimated number of confirmed COVID-19 cases March to May 2020 in pregnant women in Scotland

| | Total number of individuals testing positive (PCR) for SARS-CoV-2 (NHS labs only) | Women aged 15–44 years testing positive (PCR) for SARS-CoV-2 (NHS labs only) | Estimated number of pregnant women testing positive (PCR) for SARS-CoV-2 (NHS labs only)† |
|---|---|---|---|
| March 2020 | ≈2000 | ≈333* | ≈17 |
| April 2020 | ≈9000 | ≈1500* | ≈75 |
| May 2020 | ≈4000 | ≈667* | ≈33 |
| Total | ≈15 000 | ≈2500 | ≈125 |

*Assuming the distribution over time for this age/sex group is the same as for all tests, as age/sex breakdown only available from published information[50] for the total.
†Assuming that around 5% of the female population aged 15–44 is pregnant at any one time, and that incidence of COVID-19 is the same in pregnant and non-pregnant women.
NHS, National Health Service.

## Sample size

There are approximately 50 000 live births in Scotland per year, 13 000 terminations of pregnancy, 5000 miscarriages managed in hospital and 200 stillbirths. The estimated number of women in the population who are pregnant at any one time is approximately 42 000.

We cannot influence the number of women with confirmed, probable or possible COVID-19 available for analysis, hence sample size calculations will not be performed. We will report the precision with which we are able to estimate any association between COVID-19 and the outcomes of interest using CIs as appropriate. An approximate estimate of the expected number of confirmed COVID-19 cases in pregnant women from March to May 2020 is presented in table 2. It is likely that there will be further confirmed or probable cases in pregnant women identified through PCR testing processed through UK Government laboratories and clinical diagnoses on discharge (or possibly stillbirth or maternal death) records. In addition, it is likely that there will be considerably more possible cases among pregnant women based on the range of data sources listed above.

## ETHICS AND DISSEMINATION

COPS is a substudy of EAVE II, using unconsented data, which is covered by National Research Ethics Service Committee, South East Scotland 02 approval reference REC 12/SS/0201: SA 2 and Public Benefit and Privacy Panel approval reference 2021-0116. PHS and the Chief Medical Officer for Scotland are both (independent) data controllers for the national AAS database of termination of pregnancy notifications, thus the Chief Medical Officer has been informed of the intended use of AAS records for this study.

The results of monthly analyses summarising the incidence of COVID-19 in pregnant women, and outcomes seen in women with COVID-19 and pregnant controls, will be reported through the PHS COVID-19-enhanced surveillance cell to the Scottish Government's Chief Medical Officer's COVID-19 Advisory Group. Any results of formal modelling of outcomes that is undertaken will

be reported through the same route. Results reported through this route may be provided as management information (ie, without application of statistical disclosure control restrictions) as appropriate.

Results will also be submitted for peer-reviewed academic publication and presented at international conferences. All results put into the public domain will be subjected to statistical disclosure control according to usual PHS processes. Meta-data produced in this study will also become available to Health Data Research UK Gateway. Strengthening the Reporting of Observational Studies in Epidemiology guidance[45] and Reporting of studies Conducted using Observational Routinely-collected Data guidance[46] will be used to guide transparent reporting.

**Author affiliations**
[1]Tommy's Centre for Maternal and Fetal Health, The University of Edinburgh MRC Centre for Reproductive Health, Edinburgh, UK
[2]Usher Institute, The University of Edinburgh, Edinburgh, UK
[3]Institute of Health and Wellbeing, University of Glasgow, Glasgow, UK
[4]Public Health Scotland, Glasgow, UK
[5]School of Health, Victoria University of Wellington, Wellington, New Zealand
[6]School of Medicine, University of St Andrews, St Andrews, UK
[7]Public Health Scotland, Edinburgh, UK
[8]General Practice and Primary Care, Aberdeen University, Aberdeen, UK
[9]Department of Mathematics and Statistics, University of Strathclyde, Glasgow, UK
[10]Child Life and Health, University of Edinburgh, Edinburgh, UK

**Acknowledgements** The authors thank and acknowledge the wider EAVE II team for their support for this protocol.

**Contributors** SJS, RW, CRS and AS contributed to the conception of the study. SJS, DM, EV, CRS, HRS, UA, CM, LH, JD, LR, CR, AS and RW contributed to the study design. SJS, DM, EV, CRS, HRS, UA, CM, LH, JD, LR, CR, AS and RW authors contributed to drafting the protocol. SJS, DM, EV, CRS, HRS, UA, CM, LH, JD, LR, CR, AS and RW revised the manuscript for important intellectual content. SJS, DM, EV, CRS, HRS, UA, CM, LH, JD, LR, CR, AS and RW gave final approval of the version to be published.

**Funding** EAVE II funded by the Medical Research Council (MR/R008345/1) with the support of BREATHE - The Health Data Research Hub for Respiratory Health (MC_PC_19004), which is funded through the UK Research and Innovation Industrial Strategy Challenge Fund and delivered through Health Data Research UK. Additional support has been provided through the Scottish Government DG Health and Social Care. COPS receive additional funding from Tommy's charity (1060508; SC039280). SJS is supported by Wellcome Trust (209560/Z/17/Z)

**Competing interests** None declared.

**Patient consent for publication** Not required.

**Provenance and peer review** Not commissioned; externally peer reviewed.

**ORCID iDs**
Sarah Jane Stock http://orcid.org/0000-0003-4308-856X
David McAllister http://orcid.org/0000-0003-3550-1764
Eleftheria Vasileiou http://orcid.org/0000-0001-6850-7578
Colin R Simpson http://orcid.org/0000-0002-5194-8083
Helen R Stagg http://orcid.org/0000-0003-4022-3447
Utkarsh Agrawal http://orcid.org/0000-0001-5181-6120
Colin McCowan http://orcid.org/0000-0002-9466-833X
Leanne Hopkins http://orcid.org/0000-0002-7487-4363
Jack Donaghy http://orcid.org/0000-0002-6137-1601
Lewis Ritchie http://orcid.org/0000-0002-9380-7641
Chris Robertson http://orcid.org/0000-0001-6848-5241
Aziz Sheikh http://orcid.org/0000-0001-7022-3056
Rachael Wood http://orcid.org/0000-0003-4453-623X

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
