## [Reviewer comments · BMJ Open]

ARTICLE DETAILS

TITLE (PROVISIONAL)	COVID-19 in Pregnancy in Scotland (COPS): protocol for an observational study using linked Scottish national data
AUTHORS	Stock, Sarah; McAllister, David; Vasileiou, Eleftheria; Simpson, Colin; Stagg, Helen R.; Agrawal, Utkarsh; McCowan, Colin; Hopkins, Leanne; Donaghy, Jack; Ritchie, Lewis; Robertson, Chris; Sheikh, Aziz; Wood, Rachael

VERSION 1 – REVIEW

REVIEWER	Augusto Pereira Hospital Universitario Puerta de Hierro, Madrid, Spain
REVIEW RETURNED	12-Aug-2020

GENERAL COMMENTS	Comments to Author... This is an interesting protocol for a prospective cohort study using linked data with the aim to describe the incidence of COVID-19 in pregnancy at population level in Scotland, to determine associations between COVID-19 and adverse pregnancy, neonatal and maternal outcomes; and to establish the proportion of confirmed cases of SARS-CoV-2 infection in neonates associated with maternal COVID-19. It will be very interesting to learn about your experiences from Scotland and the data you present, will be undoubtedly of benefit to the scientific world. The methodology, concepts and the statistical study are well described in methods section. I wish you great success with your study. I have just a few comments: - Protocol paper should report planned or ongoing studies.- page 13, line 30-32: Are there any births in Scotland not controlled by the NHS (eg. Private Hospitals)?- page 12, line 44: Could you please comment on Scottish legislation? Would COVID-19 infection be included during pregnancy?- page 16, line 7: Could you please describe the medications that would be included in the protocol.- page 17, line 33-38: Would it be easier to perform the gestational calculation based on the date of the last menstrual period?- page 18, line 25-28: Should the COVID-19 be confirmed in pregnancy by a PCR, does the test has the sufficient sensitivity and specificity?
--

	- page 21, line 54-59: Where is included the arterial hypertension in this classification? - page 23, line 24-26: the author comments "We do not propose to make any formal statistical adjustment for the multiple comparisons". Why not? Are you planning to compare your results with pre-pandemic records? - page 24, line 17-31: Could you please review the trimesters of pregnancy intervals, and include a reference.
--	--

REVIEWER	Rodney McLaren Maimonides Medical Center New York, United States of America
REVIEW RETURNED	07-Sep-2020

GENERAL COMMENTS	The authors sought to primarily better delineate the effects of SARS-CoV-2 infection in pregnancy by linking population data to specific maternal and neonatal outcomes. My comments/suggestions are listed below: Line 50, page 16, how does the team plan on distinguishing a SARS-CoV-2 test for exposure or symptoms from universal testing done at a hospital? Will they be able to link hospital practices and time to the records? Line 24, page 22, What is the rationale for including possible COVID-19 under the incidence of COVID-19 as possible includes women with negative SARS-CoV-2? Including negatives will bias the results to the null hypothesis.
---

VERSION 1 – AUTHOR RESPONSE

REVIEWER ONE:

- Protocol paper should report planned or ongoing studies.

We agree. We confirm that this a study that was being planned when this manuscript was submitted, and is now ongoing (all approvals were in place allowing us to commence 25th August 2020; running to 30th September 2021). We are currently creating the cohort and linking datasets.

- page 13, line 30-32: Are there any births in Scotland not controlled by the NHS (eg. Private Hospitals)?

There are no private facilities that provide delivery care in Scotland. A small proportion ($\leq 2\%$) of births are out of hospital births (including planned home births) and may have less complete data in SMR02. However, as we are collecting data from National Records of Scotland (statutory reporting for all births regardless of setting) we will have complete coverage.

- page 12, line 44: Could you please comment on Scottish legislation? Would COVID-19 infection be included during pregnancy?

Scottish legislation mandates registration of all livebirths and all stillbirths from 24+0 weeks gestation onwards. Apologies - we are unclear on what is meant by the question "Would COVID-19 infection be included during pregnancy? However, as described in the protocol, COVID-19 status will be derived

from a variety of linked datasets including the electronic Communication of Surveillance in Scotland (ECOSS) dataset, which includes all SARS-CoV-2 test results in Scotland; and primary and secondary care health records (including maternity care and intensive care records). We expect all management of COVID-19 to be within the NHS and so we will be able to ascertain all cases of it in the population of interest.

- page 16, line 7: Could you please describe the medications that would be included in the protocol.

We will use the Prescribing Information System which holds all medications prescribed and dispensed in the community for any condition. Community prescribed medications will be used to define clinical at-risk groups. For example, we will use prescription of immunosuppressive medications such as cyclosporin and azathioprine to identify individuals at high clinical risk of COVID-19

- page 17, line 33-38: Would it be easier to perform the gestational calculation based on the date of the last menstrual period?

Gestation recorded in Scottish Morbidity Records is the 'best obstetric estimate', i.e. the date considered most likely by the treating clinician considering all available evidence. We will use this to derive the estimated date of conception. In practice the best obstetric estimate is based on first trimester ultrasound in the vast majority of women. This is the most accurate method for pregnancy dating. First trimester ultrasound is offered to all pregnant women in Scotland booking for antenatal care; all women having a termination of pregnancy; and the majority of women with any bleeding in early pregnancy. In the absence of an ultrasound the best obstetric estimate will be based on last menstrual period or clinical examination.

- page 18, line 25-28: Should the COVID-19 be confirmed in pregnancy by a PCR, does the test has the sufficient sensitivity and specificity?

Evidence regarding COVID-19 PCR test performance is maturing as the pandemic evolves. There is no reason to believe that the test will perform less well in pregnant women than in other population groups. The specificity data have been largely reassuring, with few false positives (Watson J, Whiting PF, Brush JE. Interpreting a covid-19 test result. *BMJ*. 2020 May 12;369:m1808). We can thus have a high degree of certainty that women who have a positive test have indeed had COVID-19. PCR test sensitivity has been less good, and we recognise there may be women with false negative PCR results within the cohort. Our approach of using hierarchical definitions of infection has been designed to mitigate this potential bias. As described in the protocol we will provide descriptive analyses of pregnancy outcomes not only in women with confirmed COVID-19 (who test positive on PCR), but also in women who have probable COVID -19 (which includes women with a clinical diagnosis of COVID-19 recorded in hospital admission or death records, even if PCR negative) and possible COVID-19 (which includes women who have a negative SARS-CoV-2 viral PCR test when the test was taken for clinical indications). Women with possible and probable COVID will be excluded from the comparator group to help prevent contamination of this group with women who have had false negative PCR.

- page 21, line 54-59: Where is included the arterial hypertension in this classification?

Women with hypertension are not identified as a clinical at-risk group for seasonal flu vaccination in Scotland ([https://www.sehd.scot.nhs.uk/cmo/CMO\(2019\)11.pdf](https://www.sehd.scot.nhs.uk/cmo/CMO(2019)11.pdf)), nor are they on the Scottish Shielding list (defined as very vulnerable patients; COVID-19 - <https://www.hps.scot.nhs.uk/web-resources-container/covid-19-search-criteria-for-highest-risk-patients-for-shielding/>). They are thus not included in the existing at-risk definitions on the EAVEll platform (pre-pregnancy). However, as we have stated "Other categorisation will be considered depending on numbers of pregnant women with

these conditions, and emergence of patterns of risk for COVID-19 disease.” Data on hypertension is available for all EAVEII participants. Thus, we have the capacity to explore pregnant women with chronic hypertension as an at-risk group as evidence emerges.

- page 23, line 24-26: the author comments “We do not propose to make any formal statistical adjustment for the multiple comparisons”. Why not? Are you planning to compare your results with pre-pandemic records?

We have specified a number of pregnancy outcomes which are independent from each other. Our aim is to describe outcomes, and appropriate confidence intervals will be presented. Although we have a number of outcome variables in Objective 2 there are a limited number of covariates which could be included in an investigation of confounding as we have specified a carefully matched design. Our aim in this analysis is not to test the effect of covariates but to test the effect of COVID-19 infection on outcomes so a formal use of multiple comparisons is not required over and above the reporting of effect sizes and confidence intervals. We have added text to clarify this and added more detail regarding the presentation of findings on page 21-22 as follows

P21 - “We do not propose to make any formal statistical adjustment for the multiple comparisons as the principal aim of the study is to estimate the effect of COVID-19 infection on pregnancy outcomes. The estimated effects and 95% confidence intervals will be reported for the range of outcomes. However, a caveat will be clearly expressed regarding the dangers of over interpreting these data, given the multiple outcomes used, particularly if it transpires that conflicting results are obtained from the differing outcome measures.”

P22- “We will also present our results as proportions of COVID-19 infection, together with confidence intervals based upon the Wilson method. We will describe the temporal changes in the proportion using cumulative risk models. Covariates such as the trimester of pregnancy, age of the mother and deprivation will also be included with a view to estimating the potential effects of these variables on the risk of COVID-19 infection.”

We are not planning to compare results with pre-pandemic records in the COPS study, but will do this as part of a separate protocol.

- page 24, line 17-31: Could you please review the trimesters of pregnancy intervals, and include a reference.

We have used well recognised definitions of pregnancy intervals. We have provided a reference (<https://www.acog.org/patient-resources/faqs/pregnancy/how-your-fetus-grows-during-pregnancy>) as requested.

REVIEWER TWO:

Reviewer Name: Rodney McLaren

Institution and Country: Maimonides Medical Center, New York, United States of America

Please state any competing interests or state ‘None declared’: None declared.

The authors sought to primarily better delineate the effects of SARS-CoV-2 infection in pregnancy by linking population data to specific maternal and neonatal outcomes.

My comments/suggestions are listed below:

Line 50, page 16, how does the team plan on distinguishing a SARS-CoV-2 test for exposure or

symptoms from universal testing done at a hospital? Will they be able to link hospital practices and time to the records?

Thank you. We agree that this is important and it could be a source of information bias. Early in the pandemic, testing was very restricted and confirmed to patients who were clearly symptomatic (and indeed unwell) so this is not a major issue in early analyses. However, as the pandemic has evolved, testing has become more widespread however, with both wider access to symptom-based testing in the community and more use of 'screening' type testing e.g. for patients being admitted to hospital. The indication for testing is included in the ECROSS records of SARS-CoV-2 test results. We anticipate this being of sufficient completeness to allow us to distinguish whether the test was performed for symptoms or after universal testing. If we can't distinguish between these groups, we will exclude women testing negative from both the 'case' and 'control' groups – and base our definition of possible case on presentation to various healthcare settings with relevant symptoms as described. We have added text clarifying this to the paper (page 17).

Line 24, page 22, What is the rationale for including possible COVID-19 under the incidence of COVID-19 as possible includes women with negative SARS-CoV-2? Including negatives will bias the results to the null hypothesis.

We have included hierarchical definitions of COVID-19 to mitigate against potential biases that may results from i) limitations of diagnostic strategy performance and ii) variation in availability of diagnostic tests. Definitions are very broadly based on European Centre for Disease Prevention and Control definitions (<https://www.ecdc.europa.eu/en/covid-19/surveillance/case-definition>). The definition of confirmed COVID-19 (women who test positive on PCR) may potentially bias results away from the null hypothesis (due to false negative PCR results). The definition of possible COVID-19 (due to false positive 'diagnoses') may potentially bias results towards the null hypothesis. We believe it is important to test our hypotheses in this observational cohort across the range of assumptions allowed by hierarchical definitions, including the most conservative one (i.e. possible COVID-19). We have added text clarifying this to the paper (page 18).

VERSION 2 – REVIEW

REVIEWER	Augusto Pereira Department of Obstetrics and Gynecology. Puerta de Hierro University Hospital Madrid, Spain. Autonoma University of Madrid, Spain.
REVIEW RETURNED	08-Oct-2020
GENERAL COMMENTS	Reviewer report: Most of the recommendations and suggestions have been included in the revised text. I have no further comments.